# Antibacterial Efficacy of Manuka Honey-Doped Chitosan-Gelatin Cryogel and Hydrogel Scaffolds in Reducing Infection

**DOI:** 10.3390/gels9110877

**Published:** 2023-11-06

**Authors:** Karina Mitchell, Sreejith S. Panicker, Calista L. Adler, George A. O’Toole, Katherine R. Hixon

**Affiliations:** 1Thayer School of Engineering, Dartmouth College, Hanover, NH 03755, USA; karinam25@g.ucla.edu (K.M.); sreejith.s.panicker@dartmouth.edu (S.S.P.); calista.l.adler.26@dartmouth.edu (C.L.A.); 2Geisel School of Medicine at Dartmouth, Hanover, NH 03755, USA

**Keywords:** cryogel, hydrogel, scaffolds, Manuka honey, infection, antibacterial, wound healing, tissue engineering

## Abstract

Honey has been used for centuries to reduce bacterial infection; Manuka honey (MH) possesses an additional antibacterial agent, Unique Manuka Factor (UMF). However, MH’s physical properties challenge delivery to the wound site. Tissue-engineered scaffolds (cryogels/hydrogels) provide a potential vehicle for MH delivery, but effects on bacterial clearance and biofilm formation demand further examination. MH (0, 1, 5, or 10%) was incorporated into both chitosan-gelatin (1:4 ratio; 4%) cryogels and hydrogels. To assess physical changes, all scaffolds were imaged with scanning electron microscopy and subjected to swell testing to quantify pore size and rehydration potential, respectively. As MH concentration increased, both pore size and scaffold swelling capacity decreased. Both bacterial clearance and biofilm formation were also assessed, along with cellular infiltration. Bacterial clearance testing with *S. aureus* demonstrated that MH cryogels are superior to 0% control, indicating the potential to perform well against Gram-positive bacteria. However, higher concentrations of MH resulted in cell death over time. These results support our hypothesis that MH release from 5% cryogels would induce reduced viability for four bacteria species without compromising scaffold properties. These outcomes assist in the development of a standard of practice for incorporating MH into scaffolds and the evaluation of biofilm reduction.

## 1. Introduction

For centuries, honey has been applied as a natural additive to reduce the risk of bacterial infection following traumatic injury for wound healing [1]. Honey possesses qualities conducive to preventing infection, such as high viscosity [2] and other properties including low water content, moderate acidity, high sugar content, hydrogen peroxide, and phytochemicals [1,2]. In addition, New Zealand-sourced Manuka Honey (MH) possesses a Unique Manuka Factor (UMF), which has been shown to provide bioactive agents that contribute to wound healing. The UMF rating (0–25+) of a given MH batch correlates with the content of methylglyoxal and phenol, two of the compounds responsible for MH’s antibacterial properties [1]. According to Johnston et al. [1], a honey’s UMF rating quantifies how much phenol is required to produce an antibacterial activity equivalent to honey.

As bacteria become increasingly resistant to antibiotics, additives with natural antibacterial properties, such as honey, present themselves as viable alternatives to treat Gram-positive and -negative bacterial infections. Bacteria belonging to the *Staphylococcus* genus have become increasingly resistant to antibiotic treatment; protected by a rigid peptidoglycan layer, these Gram-positive bacteria cause infection by colonizing the skin and mucosal membrane of the host [1]. *Staphylococcus* spp. can be transmitted via skin-to-skin contact or inhalation and have asymptomatically colonized up to 30% of the human population [1]. Despite these defense mechanisms, studies have shown that MH has the ability to clear wound infections colonized by *Staphylococcus* [2,3]. Another significant group of infectious Gram-positive bacteria are those of the genus *Streptococcus*. *Streptococcus* spp. induce infection by colonizing the nasopharynx and skin. Wounds, especially those associated with surgical sites, provide an optimal route of entry for this genus of bacteria; surgical-site wounds account for 25% of infections acquired in hospitals. MH effectively inhibits the establishment of *Streptococcus* biofilms and disrupts biofilms that are already established [4]. 

Gram-negative bacteria are also a significant worldwide public health problem due to their high antibiotic resistance [5]. *Escherichia coli* can cause significant ailments, including hemorrhagic colitis [6], where *E. coli* causes infection either by secreting proteins to hijack the host cell’s normal activities or by secreting toxins into the host cell [7]. This genus is commonly found on both abiotic surfaces, such as catheter implants, and biotic surfaces, including meats, vegetables, fruits, and dairy products [6]. In a study by Kim et al. [6], MH demonstrated the ability to inhibit biofilm formation by *E. coli* and to disrupt pre-existing biofilms. *Pseudomonas aeruginosa* is another Gram-negative bacterium whose growing antibiotic resistance has created a pressing need to discover alternative antimicrobial agents [8]. Found in moist habitats, *P. aeruginosa* can cause endocarditis, pneumonia, urinary tract infections, and wound infections [9]. Persons who suffer from cystic fibrosis are at high risk of deadly infections of the airway caused by *P. aeruginosa* [10]. Henriques et al. [9] deemed that MH specifically is an effective inhibitor of *P. aeruginosa*, noting that MH contributes to cell lysis and cell-surface abnormalities for this microbe.

Despite the many advantages of using MH in wound healing, the physical properties of honey (e.g., viscosity and viscidity) raise many challenges, including delivery to the wound site. Tissue-engineered scaffolds provide a biomaterial framework that allows honey to be chemically incorporated and subsequently eluted for optimal delivery with implantation; to be effective, such materials should provide structural support, facilitate tissue healing, and reduce infection risk. Hydrogel and cryogel scaffolds are two biomaterial constructs that have become popular in the field of tissue engineering. While they are chemically similar, their fabrication processes differ, leading to distinctive physical and mechanical properties. Briefly, hydrogels are crosslinked and formed at room temperature, resulting in a nanoporous structure composed of 99% water [11]. These scaffolds are characterized by their advantageous cell encapsulation/drug delivery potential; however, their low mechanical properties and nano-porosity are disadvantageous in various tissue-healing settings (e.g., bone). In contrast, when cryogel solutions are crosslinked, they are immediately placed at subzero temperatures. Freezing promotes ice-crystal formation, resulting in macropore formation throughout the scaffold. These pores are optimal for cellular infiltration, and their sponge-like structure provides mechanical durability [12]. Via the assessment of both scaffold types incorporating varying concentrations of MH, a tailorable construct can be fabricated to directly target traumatic wound healing in a variety of settings, including skin and bone.

Previous research demonstrated the incorporation of MH within hydrogel and cryogel tissue-engineered scaffolds [13,14,15,16,17,18] and their suitability as a vehicle for MH transportation and bacterial clearance. For example, Sasikala et al. [13] incorporated MH at 6 and 10% (*v*/*v*) into both chitosan-lactic acid and chitosan-acetic acid hydrogels. This study tested scaffolds against *E. coli*, *P. aeruginosa*, *S. aureus*, *Bacillus subtilis*, and the fungus *Candida albicans*, concluding that chitosan-lactic acid combined with 6% MH was effective against *S. aureus* and *E. coli.* Similarly, Bonifacio et al. [19] combined 2% MH with six types of gellan-gum-based hydrogels, which were found to be effective against *S. aureus* and *Staphylococcus epidermidis*. Our group also incorporated MH into silk-based cryogels and hydrogels at 1, 5, and 10% concentrations and evaluated these scaffolds against biofilm formation by *S. aureus* [15]. Scaffolds treated with MH showed less bacterial adhesion than the sterile control, indicating a positive effect of MH. Abd El-Malek et al. [18] incorporated MH at 20% concentration in a hydrogel (40% chitosan and 40% gelatin). The hydrogels were tested against *S. aureus*, *Streptococcus pyogenes*, *Proteus mirabilis*, *Acinetobacter baumannii*, and *P. aeruginosa.* Time-kill studies showed that the dressings incorporated with honey were successful at inhibiting all the tested bacteria for up to 12 h; MH was the most effective honey type tested. Lastly, Ullah et al. [20] used MH at 10, 15, and 20% concentrations in cellulose acetate electrospun scaffolds. These electrospun scaffolds were tested against *S. aureus* and *E. coli* and showed an acceptable amount of antimicrobial activity.

All studies have demonstrated positive results in terms of bacterial growth prevention. However, previous studies have not established an ideal honey concentration, with testing being conducted on concentrations varying from 1 to 20%. Transport and elution of honey should also be optimized at the biomaterial level. Furthermore, it is of interest to investigate the inclusion of a polymer with additional antibacterial properties. Research has demonstrated that the polymer chitosan contains natural antimicrobial properties, leading to the neutralization of negative charges on microbe surfaces and their subsequent death [21,22,23]. Therefore, the combination of this nontoxic, antimicrobial polymer with the bactericidal effects of MH could have a significant impact on biofilm formation [15].

To assist in optimizing MH scaffold parameters and delivery, this study tested both chitosan-gelatin (1:4 ratio; 4%) hydrogel and cryogel scaffolds incorporating varying concentrations of MH (0, 1, 5, or 10%) to determine the effect of honey on physical properties and antibacterial efficacy. In addition to physical and biocompatibility assessments, bacterial clearance and biofilm prevention were tested in the presence of *P. aeruginosa*, *S. aureus*, *E. coli*, and *S. sanguinis*, four prominent bacteria found in wound sites. These tests allowed us to determine the optimal scaffold type and MH concentration to maintain the desired physical properties of a traditional chitosan-gelatin hydrogel or cryogel while also displaying the ability to deter bacterial growth and/or biofilm formation.

## 2. Results and Discussion

### 2.1. Scaffold Fabrication

Both chitosan and gelatin hydrogels and cryogels were prepared as previously described [24], incorporating 0% (control), 1%, 5%, or 10% MH (MGO = 400, UMF~13; Manuka Guard, Monterey, California). These cryogels or hydrogels were used in all the experiments in this study. Visual examination showed distinct differences in the gels as a function of scaffold fabrication method and honey concentration (Figure 1).

### 2.2. Scaffold Pore Analysis

The 0% MH scaffolds served as a baseline for scanning electron microscopy (SEM) analysis, as previous studies have demonstrated efficient cellular adhesion and infiltration for chitosan-gelatin scaffolds (Figure 2). Statistical analysis found a significant increase in mean pore diameter between 0% and 1% MH cryogels and a significant decrease between 1% and 10% MH cryogels (*p* < 0.05). In a nonsignificant trend, increasing MH concentration resulted in fewer pores within the optimal range (Figure 3). The pore diameter resulting from the addition of 1% MH was comparable to 0% MH scaffolds, where only 0.21% more pores fell in the optimal diameter range. While there was no significant difference in the average pore diameter between the 1% and 5% MH scaffolds, the 5% MH had 10.41% fewer pores within the optimal pore-diameter range (*p* < 0.05). The 5% and 10% MH cryogels had similar numbers of pores in the optimal range, with only 0.15% more optimal-sized pores in the 10% MH scaffold group. It appears that the later the honey is incorporated into the scaffold solution (see Section 4), the more elongated and less circular the pores for cell infiltration. 

### 2.3. Swelling Kinetics

Swell testing was conducted on both types of scaffolds, as this property is important for clinical translation [25]. Specifically, it is desirable that scaffolds can be lyophilized for storage and rehydrated prior to use. Swell testing provides information on how long it takes a scaffold to rehydrate and what percentage of its dry mass it finally attains. With both the hydrogel and cryogel scaffolds, there was a general trend of decreasing swelling capacity as MH concentration increased (Figure 4). Statistical analysis revealed a significant difference between the swelling capacity of each scaffold at each time point in both hydrogel and cryogel samples (*p* < 0.05), with the exception of 5% and 10% hydrogel scaffolds at 2 h (*p* > 0.05). Moreover, each cryogel scaffold type differed significantly from its corresponding hydrogel scaffold type (e.g., 1% cryogel vs. 1% hydrogel). While there were mild fluctuations in values over time due to small user error, there were no significant differences between timepoints of any single group. This demonstrates that following the achievement of full swelling capacity at 2 min, this value was maintained over 24 h.

### 2.4. Mechanical Testing

Ultimate compression testing was performed to determine how MH impacts the scaffold’s physical structure. Cryogels are often used in loading environments (e.g., bone) due to their high durability and elasticity. Unamended cryogels (0% MH) served as a baseline for this analysis, as they have previously been demonstrated to be suitable for such environments [26]. Compression testing showed that the addition of MH resulted in a significantly decreased Young’s modulus as compared to the 0% MH cryogel scaffold (*p* < 0.05; Figure 5). Generally, there was also a trend that each cryogel’s modulus was higher than that of its hydrogel counterpart at a given MH concentration (e.g., 5% MH cryogel vs. 5% MH hydrogel). Note that the 10% MH scaffolds were an exception, with hydrogels having a higher modulus than cryogels. This data demonstrated that the hydrogel scaffold is typically stronger than the cryogel scaffold; however, cryogels did not exhibit crack propagation upon loading, demonstrating higher durability overall. Analysis revealed that there was no significant difference in Young’s modulus across all hydrogel scaffolds (*p* > 0.05). However, all scaffolds with MH (1%, 5%, and 10%) differed significantly from the control cryogel (0% MH). Within the cryogel samples, there was a clear trend of decreasing Young’s modulus with increasing MH concentration; at 10%, the cryogel samples had Young’s moduli similar to those of the hydrogel samples. 

### 2.5. Antibacterial Activity

As the ultimate goal is for MH to serve as an alternative method to treat infections caused by antibiotic-resistant bacteria, it is important to assess the antimicrobial efficacy of these MH scaffolds. Therefore, the bactericidal effects of both hydrogel and cryogel scaffolds on four bacterial genera were quantified. In these assays, a lawn of planktonic bacteria was spread on a rich medium, and then the materials were placed on the plate. After 24 h, the zone of clearance (indicating a lack of bacterial growth) was measured (Figure 6). Note that all scaffolds (with and without MH) displayed some degree of antibacterial activity against all bacteria tested, as indicated by their nonzero clearance zones.

*P. aeruginosa*: There was no significant difference between the hydrogel samples, nor was there a trend associated with varying MH concentration. Amongst the cryogel samples, the 0, 1, and 5% MH had similar average clearance radii, with no significant differences. However, these clearance radii were all significantly less than that for the 10% samples (*p* < 0.05). 

*S. aureus*: The hydrogel samples showed a significant increase in average clearance radius between the 1% and 5% MH scaffolds (*p* > 0.05). The average clearance radii for the 5% and 10% MH scaffolds are both higher than the 0% and 1% MH scaffolds, indicating that with higher MH concentration, scaffolds will have a larger average clearance radius. The cryogel samples showed no significant difference among MH concentrations; however, an increase in MH concentration shows a trending increase in average clearance radius.

*S. sanguinis*: Both cryogel and hydrogel samples showed no significant differences with varying MH concentrations. For the hydrogel samples, the 0% and 1% MH concentrations had a lower average clearance radius than the 5% and 10% MH concentrations, but none of these changes was statistically significant (*p* > 0.05). Varying the MH concentrations for the cryogel samples did not correlate with an increase or decrease in the average clearance radius.

### 2.6. Biofilm Formation

In these assays, the antimicrobial and antibiofilm activity of scaffolds versus biofilm-forming bacteria was measured. The gel samples were incubated in the wells of a 12-well plate inoculated with the indicated bacteria. We determined the antimicrobial activity versus planktonic bacteria by measuring absorbance at 550 nm. We determined the extent of biofilm formation by staining surface-attached bacteria with the dye crystal violet and quantifying the extent of staining.

*E. coli*: The hydrogel samples showed no significant difference in planktonic bacterial growth across MH concentrations (Figure 7); however, there was a significant increase in biofilm formed between the 0% and 10% MH scaffolds (*p* > 0.05). For the cryogel samples, there was a significant decrease in planktonic bacterial growth between the 0% and 5% MH scaffolds (*p* > 0.05). However, for the cryogel samples, there was no significant difference or trend in biofilm formation across all MH concentrations. 

*P. aeruginosa*: The hydrogel scaffolds showed a trend in increased planktonic bacterial growth as MH concentration was increased. Specifically, the hydrogel samples showed a significant increase in planktonic bacteria growth between the 0% and 5% MH, 0% and 10% MH, 1% and 5% MH, and 1% and 10% MH hydrogels. Despite this observation, there was no significant difference in biofilm formation, but there was a general decrease in biofilm formation as MH concentration increased. The cryogel samples showed a general increase in planktonic bacterial growth as MH concentration increased, similar to that of the hydrogel samples (Figure 8). There was a significant increase in planktonic bacterial growth between the 0% and 10% MH, the 1% and 5% MH, and the 1% and 10% MH cryogels (*p* > 0.05). While the higher concentrations promoted planktonic bacteria, there was a general decrease in biofilm formation as MH concentration increased for the cryogel scaffolds. There was a significant decrease in biofilm formation between the 1% and 5% MH cryogels and between the 1% and 10% MH cryogels. An outlier to this general trend was the significant increase in biofilm formation between the 0% and 1% MH scaffolds. 

*S. aureus*: The hydrogel scaffolds displayed no significant difference in planktonic bacterial growth; however, there was a general trend of decreasing planktonic bacterial growth as MH concentration increased. There was no significant difference nor a general trend across MH concentrations in terms of biofilm formation. Comparatively, the cryogel samples showed a general decrease in planktonic bacterial growth as MH concentration increased, with a significant decrease between the 0% and 5% MH as well as the 0% and 10% MH cryogels (*p* < 0.05; Figure 9). There was no significant difference nor a general trend in terms of biofilm formation across all MH concentrations.

*S. sanguinis*: The hydrogel samples showed no significant difference in either planktonic bacterial growth or biofilm formation but demonstrated a general decreasing trend in both phenotypes as MH concentration was increased. Similar to the hydrogels, the cryogel samples showed no significant difference in terms of planktonic bacterial growth; however, there was a general trend of decreasing planktonic growth as MH concentration increased (Figure 10). The cryogel samples showed no significant difference in biofilm formation; however, there was also a general decrease in biofilm formation as MH concentration increased.

### 2.7. Cell Infiltration

Scaffolds are intended to assist with the regeneration of tissues and, therefore, must both encourage and support cell infiltration and matrix deposition. In the case of hydrogels, cellular incorporation is completed via cell encapsulation during the initial scaffold formation; the nanoporous structure of hydrogels does not allow for infiltration to occur following scaffold formation. Cryogels, on the other hand, allow for cell incorporation via a seeding process following the formation of the scaffold, where the macroporous structure supports cell migration and infiltration throughout the porous structure. Cellular infiltration analysis provides necessary information on the environment’s long-term support of cellular infiltration, proliferation, and survival [27]. 

Cryogel scaffold samples were seeded with human osteosarcoma (MG-63) cells and cultured for 7, 28, and 35 days (see Section 4 for details). In this study, all cryogel scaffolds were shown to provide a conducive environment for cell infiltration, as demonstrated by the presence of cells at day 7 (Figure 11). On Day 28, there were cells present in all scaffolds, with the exception of the 10% MH scaffolds, which could be a product of the cytotoxicity of the honey. This trend continued, as cells were only observed in the 1% MH scaffolds (images were not obtained for the 0% MH scaffolds) at Day 35, with no cells within the 5% and 10% MH scaffolds (Figure 12). 

### 2.8. Discussion

This study focused on using chitosan-gelatin hydrogel and cryogel scaffolds to provide a vehicle for MH delivery while assessing *Escherichia coli*, *Pseudomonas aeruginosa*, *Staphylococcus aureus*, and *Streptococcus sanguinis* clearance and biofilm formation. Future work should measure the polydispersity of the chitosan, which provides an additional antibacterial agent to the scaffold constructs [21,22,24].

SEM analysis of all scaffolds provided insight into the effects of MH on pore structure and size (Figure 2). These data showed that as MH concentration increased, the integrity of pore structure decreased. Cryogel pores lost the circular structure required for cell infiltration and, instead, displayed angular pore geometry, fewer pores, and a larger number of pore sizes falling outside of the optimal range for cell infiltration. During fabrication, the MH was incorporated into the solution prior to crosslinking. Therefore, following crosslinking and freezing, the MH is directly incorporated within the polymer chain struts. This results in multiple changes in both the ice crystal formation and shape of the polymer chains forming the pores, directly leading to decreased integrity of the pore structure. Additionally, the hydrophilic/hydroscopic nature of the honey directly pulls and attracts the water with the distribution of honey throughout the polymer struts. As the polymer chains are disrupted by the honey, this results in an angular geometry and generally smaller pore sizes [28,29,30].

Swelling kinetics are vital for understanding rehydration potential, as well as nutrient and waste movement through scaffolds in vivo (Figure 4). This study supports previous work showing that swelling capacity decreases with increasing MH concentration [17]. This is most likely due to MH’s impact on pore formation via its integration within polymer chains during crosslinking. As SEM imaging and analysis of cryogels revealed that pore size generally decreased with increased MH concentration, this notion serves as a viable explanation for why the swelling capacity also decreased with increasing amounts of MH. Hydrogels exhibited a similar trend of decreased swelling with increased MH concentration. Here, too, it is plausible that MH is being incorporated within the polymer chains, disrupting swelling potential. Note that hydrogels exhibited reduced swelling compared to their cryogel counterparts due to the sponginess of the cryogels and the ability to rehydrate to a higher capacity. 

In addition to swelling, the effect of MH on mechanical integrity is also important for understanding in vivo applicability for tissue regeneration and wound healing (Figure 5). Compression data showed that increased MH concentrations decreased the overall strength of the cryogel, which again could be attributed to the impact that the honey has on pore formation and polymer chain distribution. In contrast, hydrogel samples showed no significant differences or an overall change in Young’s modulus with changes in MH concentration. Hydrogels are not macroporous, and thus, their nanoporous, water-filled structure directly elucidates why they are generally weaker and not conducive for supporting exterior loads. Despite these differences, hydrogels have been shown to be beneficial in many tissue-engineering and wound-healing applications (e.g., wound healing), where the lack of mechanical changes with increased MH concentrations could prove advantageous in additive delivery [31].

Following structural characterization, we assessed the potential of chitosan-gelatin scaffolds for antibacterial activity (Figure 6). Interestingly, all scaffolds without MH had some antibacterial properties, likely due to the natural antibacterial properties of chitosan [21,22,32]. Overall, all scaffolds (with and without MH) showed at least some antibacterial activity against all of the bacteria tested here. In general, both cryogels and hydrogels had the highest antibacterial effect against *E. coli* and the lowest antibacterial effect against *P. aeruginosa.* Across all bacterial strains, the cryogel scaffolds generally had higher average clearance radii in comparison to their hydrogel counterparts. We propose that this finding is most likely due to cryogels’ pore structure and polymer chain distribution leading to a stronger release of MH. 

Biofilm formation is also an important component in wound healing and infection management (Figure 7, Figure 8, Figure 9 and Figure 10). In this study, biofilm assays demonstrated that both cryogels and hydrogels, with and without MH, are capable of biofilm inhibition. The *E. coli*, *P. aeruginosa*, and *S. aureus* assays showed that hydrogels generally had higher average anti-biofilm activity than cryogels. The opposite is true of the assays conducted against *S. sanguinis*, with the cryogel samples performing better than their hydrogel counterparts. Overall, both cryogel and hydrogel samples were most effective against *S. aureus* and least effective against *S. sanguinis*. It is particularly interesting that the scaffolds were least effective against *S. sanguinis* in the biofilm assay and arguably performed second best against this bacterium in the antibacterial testing.

Finally, cell infiltration is necessary to assess a scaffold’s ability to support new tissue, which is also affected by (potential) cytotoxic effects of MH [33,34,35,36]. In this study, cell infiltration studies showed that plain cryogel scaffolds (with or without MH) provided a biocompatible environment for up to 21 days (Figure 11). However, on Day 28, we saw that there were no longer cells present in the 10% MH scaffolds only. When the cell study was extended to Day 35 (Figure 12), cells were no longer present in the 5% MH scaffolds. Previous studies indicate that the longer human cell lines are exposed to MH, the more cytotoxic they become to them [34]. It remains unclear if this effect would also occur in vivo, as this study only examined the release of MH in a contained well plate and did not account for fluid flow and the dynamic environment in the body. Thus, further studies should be conducted to analyze both the effect of fluid flow and the degradation of the scaffolds as related to potential cytotoxicity. An additional study that could prove informative includes the quantification of glucose release from the MH overtime to assess cellular exposure; however, similar studies have already demonstrated release over 14 and 28 days in both silk fibroin and chitosan-gelatin cryogels [17,36]. Notably, one of these studies tested two different MH UMFs (5 and 20), where there was no significant difference between any of the scaffolds, regardless of which UMF was incorporated. This supports an extended release where UMF value has no effect on clearance over time as the rate of elution is gradual. From these studies and matching data for pore size, bacterial clearance, etc., demonstrated in this current manuscript, the same glucose release can be anticipated in this study. Instead, this manuscript uniquely demonstrates a wider range of bacterial clearance testing, as well as the addition of biofilm formation assessment.

## 3. Conclusions

In conclusion, this study focused on testing varying concentrations of MH (0, 1, 5, or 10%) in hydrogel and cryogel scaffolds. Scaffolds were evaluated to understand MH’s effects on physical properties and its antibacterial activity. Results indicate that MH can be incorporated into both hydrogel and cryogel scaffolds with varying effects on bacterial clearance and biofilm formation. Specifically, analysis of all scaffolds demonstrated that as MH concentration increased, the integrity of pore structure and scaffold swelling capacity also decreased. This is most likely due to the MH being incorporated within the polymer chains and disrupting the physical scaffold properties. This was further supported by a decreased mechanical strength with increased concentrations of MH. Despite this variability in scaffold properties, the addition of MH had at least some antibacterial activity against all of the Gram-positive and -negative bacteria tested. This was further supported by biofilm assays where biofilm inhibition was also noted. Despite the advantages of the MH, cell infiltration was affected over time with increasing MH concentrations, where 10% MH proved cytotoxic as early as 28 days. This demonstrates a required balance that must be achieved between necessary physical properties, anti-bacterial/-biofilm formation, and cellular compatibility. Notably, by using a chitosan–gelatin material combination, an additional antibacterial component from the chitosan is provided, leading to some clearance even without the addition of MH. Overall, the 5% MH cryogels and hydrogels appear to withstand physical property changes, maintaining desired porosity, swelling, and mechanics while manifesting antibacterial and anti-biofilm activity. Future studies should investigate how 5% MH cryogel and hydrogel scaffolds perform in dynamic environments (e.g., fluid flow and mechanical loading) they would typically be exposed to in vivo (for bone, muscle, skin, etc.).

## 4. Materials and Methods

### 4.1. Scaffold Fabrication Optimization

To create the solutions for plain hydrogel and cryogels, 10 mL of DI water was combined with 1% acetic acid (Thermo Fisher Scientific, Waltham, MA, USA). Following this, the solution was split into 8- and 2-mL aliquots, where 80 mg of chitosan (Sigma-Aldrich, St. Louis, MO, USA) was added to the larger aliquot, and this solution was spun on a mechanical spinner for 30 min. Note that chitosan’s average molecular weight is 50,000–190,000 Da and 75–85% deacetylated, with a viscosity of 20–300 cP.

After complete dissolution, 320 mg of gelatin (Sigma-Aldrich, St. Louis, MO, USA) was added, and the solution spun on the mechanical spinner for another 30 min. Subsequently, 1% glutaraldehyde (Sigma-Aldrich, St. Louis, MO, USA) was added to the 2 mL aliquot to prepare the crosslinker. After both aliquot solutions were fully mixed, hydrogel or cryogels could be formed. To form hydrogels, the two solutions were combined to initiate crosslinking and poured into a sealed, cylindrical mold at room temperature. In comparison, cryogels were formed by allowing the 8-and 2-mL solutions to cool at 4 °C for one hour. The mold that the cryogel solution was poured into was pre-cooled to −20 °C. After the cryogel solution and crosslinker solution were cooled, they were decanted into each other, poured into the pre-frozen mold, and frozen at −20 °C for 24 h. The scaffolds were imaged using an SEM (Tescan Vega3 SEM; Tescan, BRNO, Brno, Czech Republic) to evaluate pore size and structure. To initially determine the best method to incorporate MH into the 4%-chitosan-gelatin scaffolds, MH was added to the scaffolds at different stages using four mixing methods. Note that the concentration of MH used (0, 1, 5, or 10%) was based on previous work by our group and others, demonstrating the advantages of these different concentrations related to bacterial clearance and scaffold properties [17,19,20,37].

The four methods of fabrication evaluated were to incorporate the MH (1) using the sonicator bath, (2) the mechanical spinner, (3) added with the chitosan, and (4) added with the gelatin. All scaffolds were lyophilized and imaged using SEM (Figure 2). Analysis was performed to quantify pore size, a good indicator of how conducive a scaffold is for cellular adhesion and proliferation. The optimal pore diameter is generally 100–200 µm [38]. Pore analysis enables the calculation of the number of pores with diameters in this range. This is an important characteristic for assessing mechanical properties such as compressive strength and swelling abilities, as well as cellular adhesion. Specifically, scaffold pores must be of a certain shape and size for cells to infiltrate, adhere, and proliferate to survive. Note that a sterile syringe was used to add the MH to ensure proper volume for a 1%, 5%, or 10% MH cryogel. For the first method, MH was added to the 8 mL, 1% acetic acid aliquot and mixed until fully homogeneous with an acetic acid solution using a sonicator bath (Branson/Emerson Electric, Brookfield, CT, USA). The second method also incorporated MH into the 8 mL 1% acetic acid mixture, but this aliquot was mixed on the mechanical spinner to achieve dissolution. For method three, the MH was added simultaneously with the chitosan, and the remainder of the protocol was followed as described previously. For method four, the MH was added simultaneously with the gelatin prior to mixing on the mechanical wheel. The first method of fabrication (sonication in 1% acetic acid) was determined to be the optimal method for incorporation, verified via SEM (Figure 13). Specifically, pores were rounded for cellular integration and more homogeneously shaped compared to the other methods; this is how all scaffolds were fabricated for the remainder of the tests. 

### 4.2. Cryogel Pore Assessment

To conduct pore analysis, hydrogel and cryogel scaffolds (with or without MH) were imaged. Prior to imaging, all the scaffolds were frozen at −80 °C for 1 h and lyophilized (Labconco) for 24 h. The samples were then mounted on an aluminum stub and sputter coated (HUMMER 6.2; Anatech, Sparks, NV, USA) for 240 s in gold at 15 mA under the pulse setting to avoid overheating. All images were taken at a 30-kV beam intensity. Images at 100× were used to conduct pore analysis. Cryogel images were run on a Python program designed to calculate average pore diameter. Hydrogels were imaged but not quantified due to their lack of macroporous structure. 

### 4.3. Swell Testing

Swell testing was completed to evaluate the swelling capacity of all of the scaffolds, with and without the addition of MH. Lyophilized samples from each scaffold type (*n* = 3) were used to collect data. The original dry weight of each scaffold was first recorded, and then each scaffold was placed in weighboats filled with deionized (DI) water (enough to fully submerge the samples; ~5 mL) to reach full rehydration. The weight of each individual scaffold was measured at the following time points: 2 min, 4, 10, 20, 40 min, 1, 2, 4, and 24 h. To calculate swelling capacity, dry weight was subtracted from the hydrated weight. This total was divided by the dry weight and multiplied by 100 to determine the % swelling capacity [24].

### 4.4. Mechanical Testing

Mechanical testing was performed to evaluate the compressive strength of both hydrogel and cryogel scaffolds (with and without MH; *n* = 3). For this test, it was imperative that the scaffolds be uniform in shape and size, able to stand on their own, and large enough to test using the Instron 6800 series. To achieve this shape, a silicone cube-shaped mold was used, with dimensions of ~0.88 cm^3^. Prior to placing the samples on the Instron, the length, width, and height of each were confirmed using dial calipers. This information was programmed into the Instron, and the machine was calibrated to compress each sample to 80% of its original height. The stress and strain data were then used to calculate the Young’s modulus for each sample and averaged across groups. 

### 4.5. Antibacterial Assay

Bacterial testing was conducted to analyze the bactericidal effects of MH on four genera of bacteria: *E. coli*, *P. aeruginosa*, *S. aureus*, and *S. sanguinis*. The bacterial cultures were grown overnight in lysogeny broth (LB) and then diluted in LB, 1 to 1000, such that 1 µL of bacteria (~1 × 10^9^ CFU/mL) was inoculated into 1 mL of LB. This bacterial suspension was vortexed, and 100 µL of the bacterial dilution was added to the surface of an agar plate, and then evenly spread using a cell spreader and allowed to dry for 10 min. Once dried, 5 mm diameter scaffolds (*n* = 3) were placed on the plate in a triangular fashion with enough room separating them for zones of clearance to be distinguishable. The plates were incubated overnight (~24 h) at 37 °C. Once removed from the incubator, images were obtained of each plate and analyzed using ImageJ (NIH). Bacterial clearance zones (an indication of lack of bacterial growth) were calculated by determining the average diameter of the scaffold (determined by three different measurements of scaffold diameter), subtracting this value from the average diameter of the zone of clearance ring (determined by three different measurements of the clearance ring diameter), and then dividing this in half to yield the average clearance radius (cm). Each experiment was repeated in triplicate. For each experiment, a fresh, independent culture was used.

### 4.6. Biofilm Formation Assay

Biofilm assays were performed to evaluate the effectiveness of each scaffold type in disrupting the formation of biofilms by *E. coli*, *P. aeruginosa*, *S. aureus*, and *S. sanguinis* [39]. Three samples of each scaffold type were used in each replicate, and a total of three experiments were performed per bacterial type. For each experiment, a fresh, independent culture of bacteria was used. The hydrogel and cryogel samples were extracted from their original syringe mold using a 5 mm biopsy punch, and each punch was cut in half longitudinally (~5 mm). Each sample was submerged in 2 mL of a 25 mL inoculated medium in a 12-well plate. For *P. aeruginosa*, the culture medium consisted of 25 mL of 1 × M63, 25 µL of 1 M MgSO_4_, 500 µL of 20% arginine, and 500 µL of overnight, LB-grown *P. aeruginosa* culture. For *E. coli*, the 25 mL culture medium consisted of 25 mL of LB, 25 µL of 1 M MgSO_4_, 1.25 mL of 20% CAA, 500 µL of 20% glucose, and 500 µL of LB-grown *E. coli* culture. For *S. aureus*, the mL culture medium consisted of 25 mL of tryptic soy broth, 25 µL of 1 M MgSO_4_, and 500 µL of an *S. aureus* culture grown in tryptic soy broth overnight. For *S. sanguinis*, the culture medium consisted of 25 mL of THY (Todd-Hewitt broth w/2% yeast extract), 25 µL of 1 M MgSO_4_, and 500 µL of overnight, THY-grown *S. sanguinis* culture. The experiments were incubated for 24 h at 37 °C. At the 24 h mark, 100 µL of the medium was taken from each well plate containing medium/sample, and the absorbance was measured using a spectrophotometer (550 nm; M2 Spectra Max Multilabel Microplate Reader, Molecular Devices, LLC, San Jose, CA, USA). The well plates were then rinsed by submerging them in water and decanting the liquid; 2 mL of crystal violet (0.1% *w*/*v* in water) was used as an indicator for the presence of a biofilm [40]. Following previously reported protocols [40], the plate with crystal violet was incubated for 5 min, after which the plates were thoroughly rinsed by submerging the well plate in water and air-dried overnight. The following day, 2 mL of 30% glacial acetic acid was added to each well and plated on a rocker for 5 min. Next, 100 µL was removed from each well and added to a 96-well plate, and the absorbance was measured using a spectrophotometer at 550 nm with 30% glacial acetic acid as the blank. This is based on standard of practice [41,42,43,44].

### 4.7. Cell Infiltration

Cell infiltration studies were administered to determine the capacity for cells to adhere to and survive within the cryogel scaffolds over time [45,46]. Cryogels incorporated with 0%, 1%, 5%, and 10% MH were formed in 3 mL syringes, removed, and sterilized in 1% peracetic acid for 90 min followed by three consecutive 10 min washes in sterile 1× phosphate-buffered-saline (PBS). After sterilization, each scaffold was seeded with 50,000 human osteosarcoma (MG-63, passage 92; ATCC) cells by slowly dripping 100 µL of cell suspension on the top of the scaffold in individual wells of a 24-well plate. Complete media was composed of Dulbecco’s Modified Eagle Medium supplemented with 4.5 g/L glucose, L-glutamine, and 110 mg/L sodium pyruvate (Gibco, Waltham, MA, USA), 10% fetal bovine serum (Omega Scientific, Tarzana, CA, USA), and 1% penicillin-streptomycin solution (Gibco). Cryogels were incubated for one hour at 37 °C and 5% CO_2_ to allow for cellular attachment. Thereafter, 1.5 mL of media was added, so scaffolds were submerged, and well plates were placed back in the incubator. Media were replaced every two to three days, and cryogels were removed on days 7, 14, 21, 28, and 35. Upon removal, samples were placed in formalin for 24 h for preservation and subsequently cryoprotected with 30% (*w*/*v*) sucrose (Sigma-Aldrich) for 24 h. To embed the scaffolds, a 5% (*w*/*w*) porcine gelatin (Sigma-Aldrich) and 5% (*w*/*w*) sucrose solution was dissolved in deionized water and pre-warmed to 45 °C. Scaffolds were then placed in molds, and the embedding solution was pipetted into the molds. Scaffolds were incubated for 2 h at 45 °C. After incubation, molds were subsequently immersed in an acetone and dry ice bath until frozen. The samples were transferred to a −80 °C freezer until they were ready to be sectioned. Cryosectioning was completed at −20 °C (16–20 µm sections). Sections were stained with 4′,6-diamidino-2-phenylindole, dihydrochloride (DAPI; BD Biosciences, Franklin Lakes, NJ, USA). Images were taken by an inverted compound light microscope (Laxco SLi3PRO Inverted Fluorescence Microscope) at 4×, 10×, and 20× lens magnification. 

### 4.8. Statistical Analysis

All data were analyzed using GraphPad Prism software. The data were analyzed via ordinary one-way ANOVA, adjusted for multiple comparisons test (compares the mean of each column with the mean of every other column). Significance was determined at *p* < 0.05. 

## Figures and Tables

**Figure 1 gels-09-00877-f001:**
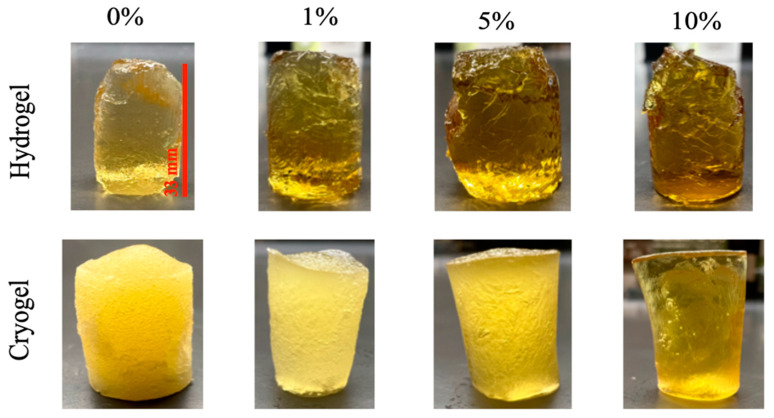
All scaffolds examined in this study, formulated as described. Hydrogels (**top row**) and cryogels (**bottom row**) are arranged as a function of percent MH incorporated.

**Figure 2 gels-09-00877-f002:**
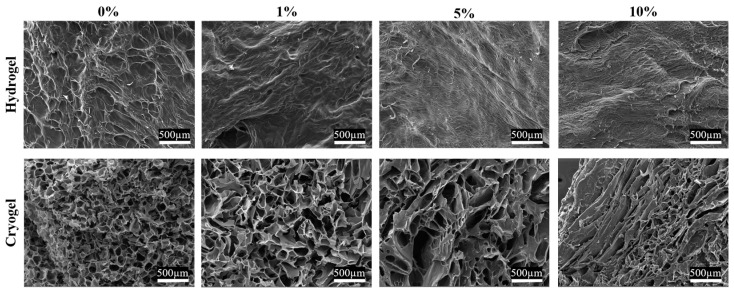
SEM images of all the scaffolds used during this study. Images of cryogel samples were used to conduct pore analysis; hydrogels did not exhibit pores on the macro-scale (nano) and thus could not be analyzed. White scale bar is 500 µm.

**Figure 3 gels-09-00877-f003:**
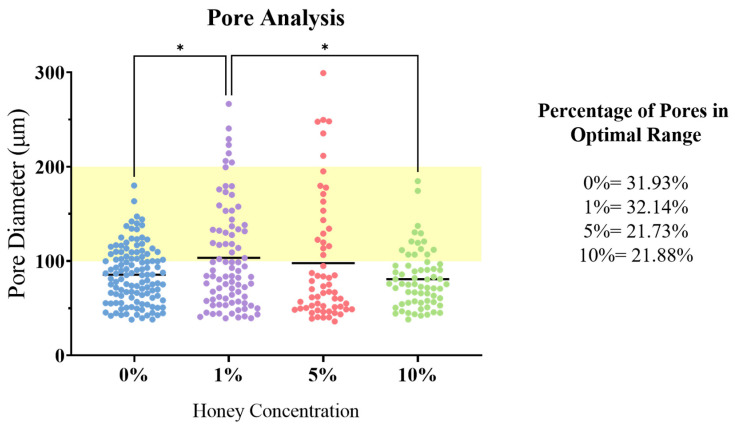
Pore distribution with increasing concentration of MH in cryogel scaffolds. Shaded yellow depicts optimal pore range (100–200 µm) for cell infiltration and vascular formation. (* *p* < 0.05).

**Figure 4 gels-09-00877-f004:**
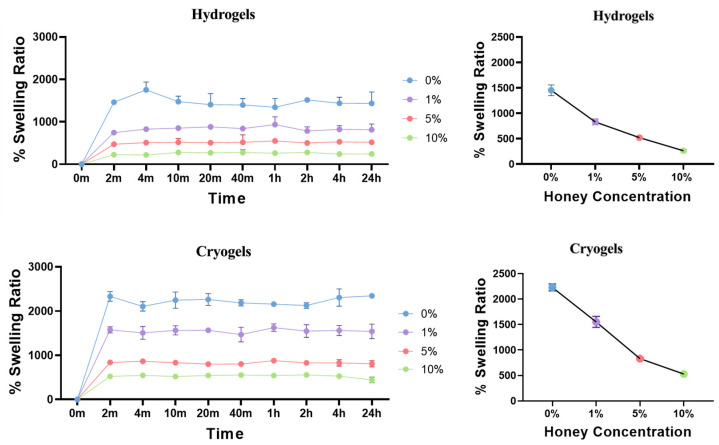
Hydrogel and cryogel swelling ratios over 24 h. Linear regression models showed that as MH increased, scaffold swelling potential decreased (*p* < 0.05).

**Figure 5 gels-09-00877-f005:**
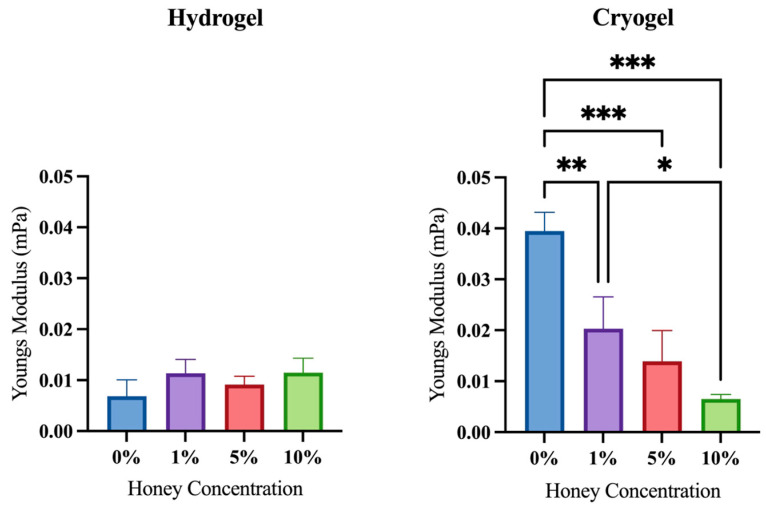
Ultimate compression of cryogel and hydrogel scaffolds. Increasing concentrations of MH in cryogels resulted in significantly lower Young’s modulus (* *p* < 0.05, ** *p* < 0.05, *** *p* < 0.01).

**Figure 6 gels-09-00877-f006:**
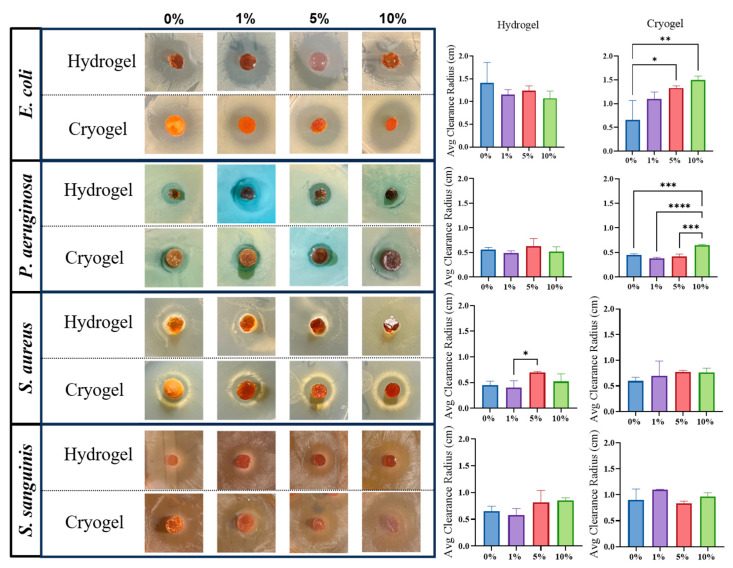
Visual and statistical results from bacterial clearance testing. All scaffolds are 5 mm in diameter. (* *p* < 0.05, ** *p* < 0.05, *** *p* < 0.01) *E. coli*: For the hydrogel samples, while there were no significant differences between any of the samples, the average clearance radius generally decreased with the incorporation of MH. The cryogel samples showed a trend of increasing average clearance radius with increased MH concentration. The 5% and 10% MH cryogels showed a significant increase in average clearance radius in comparison to the control (0% MH) cryogel (**** *p* < 0.05).

**Figure 7 gels-09-00877-f007:**
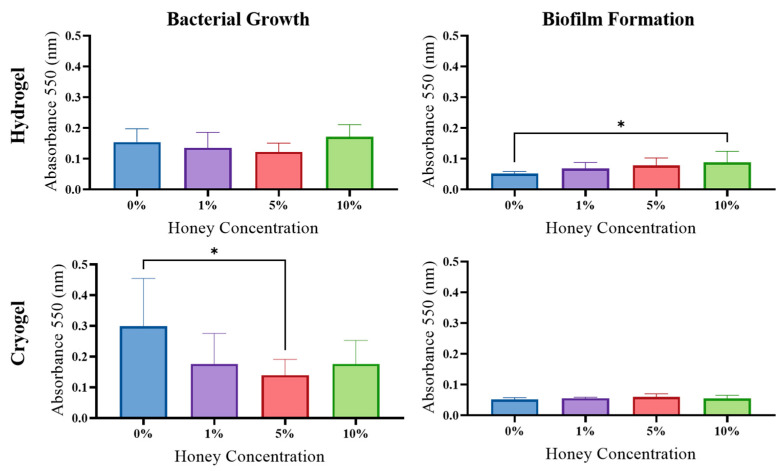
*E. coli* results from growth and biofilm analysis. (* *p* < 0.05), one-way ANOVA.

**Figure 8 gels-09-00877-f008:**
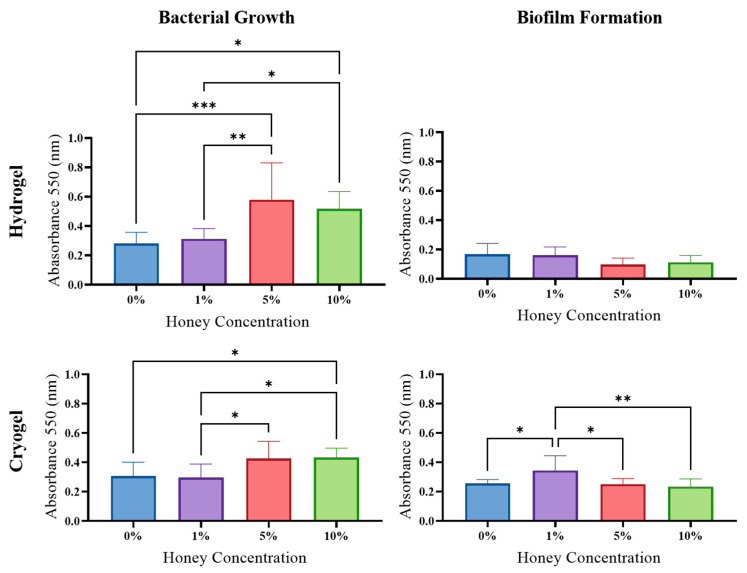
*P. aeruginosa* results from growth and biofilm analysis (* *p* < 0.05, ** *p* < 0.05, *** *p* < 0.01), one-way ANOVA.

**Figure 9 gels-09-00877-f009:**
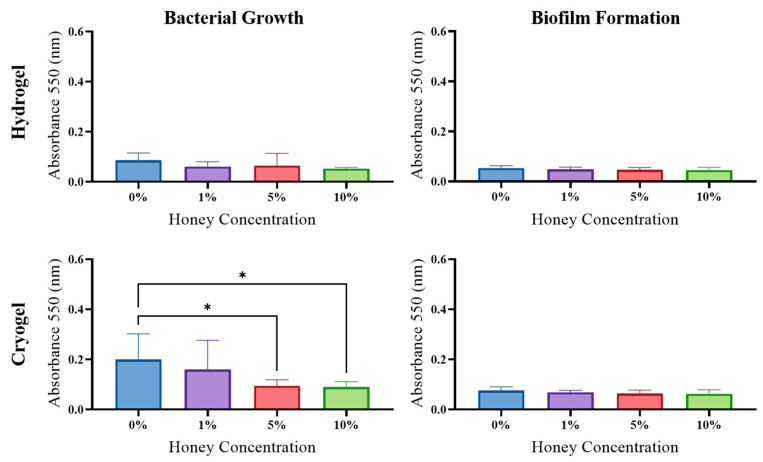
*S. aureus* results from growth and biofilm analysis (* *p* < 0.05), one-way ANOVA.

**Figure 10 gels-09-00877-f010:**
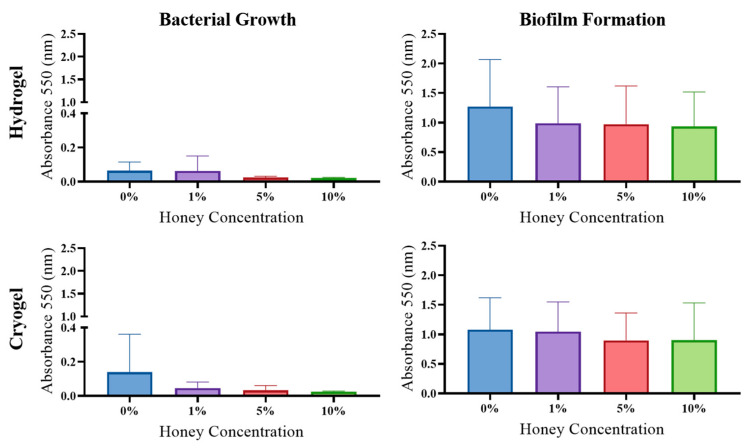
*S. sanguinis* results from growth and biofilm analysis. No significance was noted (*p* > 0.05), one-way ANOVA.

**Figure 11 gels-09-00877-f011:**
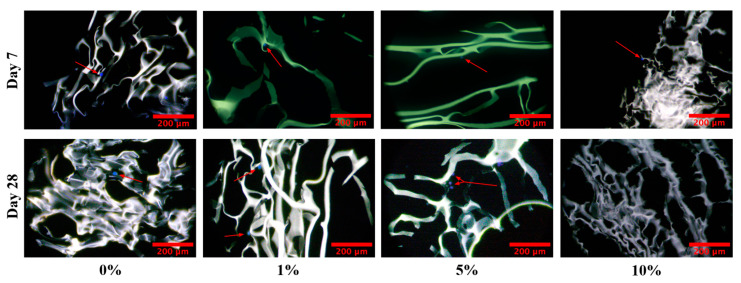
Day 7 and 28 images from cell infiltration analysis. Cells are colored blue/green (DAPI stain) and identified with red arrows. Note that there are no cells present on the 10% MH scaffolds on Day 28. Red scale bar denotes 200 µm.

**Figure 12 gels-09-00877-f012:**
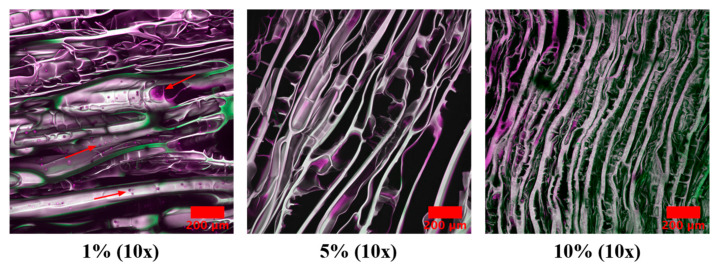
Day 35 images of 1%, 5%, and 10% MH scaffolds. Cells are colored pink (only seen in 1% MH), where images are false colored (purple/green) and overlaid to identify the scaffold. Cells in the 1% MH samples are identified with red arrows. Note that there are no cells present on the 5% and 10% MH scaffolds. Red scale bar denotes 200 µm.

**Figure 13 gels-09-00877-f013:**
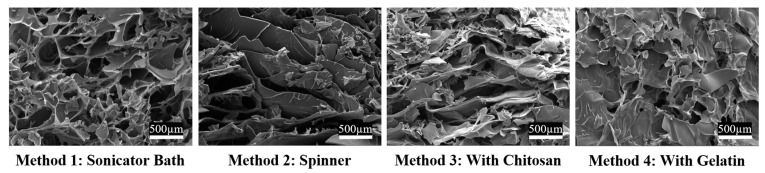
SEM images depicting scaffolds at the varying steps where MH was added to the original hydrogel/cryogel solution. Method 1: MH added prior to chitosan and mixed using sonicator bath. Method 2: MH added prior to chitosan and mixed using the mechanical spinner. Method 3: MH added with chitosan and mixed using the mechanical spinner wheel. Method 4: MH added with gelatin and mixed using the mechanical spinner wheel. White scale bars are 500 µm.

## Data Availability

The data presented in this study are available upon request from the corresponding author.

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
