# Peer review of "Antibacterial Efficacy of Manuka Honey-Doped Chitosan-Gelatin Cryogel and Hydrogel Scaffolds in Reducing Infection"

_gels, 2023, doi:10.3390/gels9110877_

Round 1

Reviewer 1 Report

Comments and Suggestions for Authors

In this study, authors focused on testing varying concentrations of MH in hydrogel and cryogel scaffolds. Results indicate that MH can be incorporated 385 into both hydrogel and cryogel scaffolds with varying effects on bacterial clearance and biofilm formation. Future studies should investigate how 5% MH cryogel and hydrogel scaffolds perform in the dynamic environments (e.g., fluid flow and mechanical loading) they would typically be exposed to in vivo for bone, muscle, skin, etc.). The following issues should be considered during the revision.

1.      In figure 6, a scale bar or even a Centimeter-ruler is needed to reflect these two livers in the same magnification. In many figures, their panels should be labelled as a,b,c…, so as to make their sense more clear.

2.      In figure 4, authors showed the hydrogel and cryogel swelling ratios over 24 h. However, in some time points, the swelling ratios were decreased. Why? Please give the discussions.

3.      The conclusion have 8 paragraphs. I suggest authors to divide them into two parts, part I (para 1-7 as the discussion) and part II as the conclusion. There are too many keywords, should be simplified.

4.      Data in figure 2 showed that as MH concentration increased, the integrity of pore structure decreased, why? What is the underlying reasons for that?

5.      Also, why cryogel pores lost the circular structure required for cell infiltration and, instead, displayed angular pore geometry, fewer pores, and a larger number of pore sizes falling outside of the optimal range for cell infiltration?

6.      In figure 7-10, authors using the absorbance 550 nm to indicate the bacterial growth and biofilm formation, which is not intuitionistic to illustrate the antibacterial properties of the hydrogels and cyrogels. Should they changed into antibacterial rations?

7.      Several related papers on this area can be included: https://doi.org/10.3390/gels5020021; https://doi.org/10.1016/j.ijbiomac.2023.125754.

Comments on the Quality of English Language

No further comments.

Author Response

Dear Editors and Reviewers,

We are pleased to submit the revised version of Gels (ID# gels-2652916): “Antibacterial Efficacy of Manuka Honey-Doped Chitosan-Gelatin Cryogel and Hydrogel Scaffolds in Reducing Infection.” We appreciate the reviewers' valuable feedback and have thoroughly addressed each of their concerns as detailed below. All revisions to the manuscript are tracked changes. Additionally, we have included a clean version of the manuscript with all changes accepted. We enhanced both the text and figures to enrich the paper; these modifications have notably enhanced the paper and improved its clarity.

Reviewer 1

In this study, authors focused on testing varying concentrations of MH in hydrogel and cryogel scaffolds. Results indicate that MH can be incorporated 385 into both hydrogel and cryogel scaffolds with varying effects on bacterial clearance and biofilm formation. Future studies should investigate how 5% MH cryogel and hydrogel scaffolds perform in the dynamic environments (e.g., fluid flow and mechanical loading) they would typically be exposed to in vivo for bone, muscle, skin, etc.). The following issues should be considered during the revision.

  1. In figure 6, a scale bar or even a Centimeter-ruler is needed to reflect these two livers in the same magnification. In many figures, their panels should be labelled as a,b,c…, so as to make their sense more clear.

All scaffolds from this picture are 5-mm in diameter, which has now been noted within the methods and figure caption. Related to panels, we appreciate the reviewer noticing this and have modified the figure accordingly. Instead of adding “a,b,c,…” designations to the figure, we added labels to the top of the columns for clarification of honey %.

  1. In figure 4, authors showed the hydrogel and cryogel swelling ratios over 24 h. However, in some time points, the swelling ratios were decreased. Why? Please give the discussions.

While there is some fluctuation in swell values, this is just due to slight user error between samples where no sample type had significant differences between 2 and 24 hours. We have added clarification regarding this in both the results and discussion section.

  1. The conclusion have 8 paragraphs. I suggest authors to divide them into two parts, part I (para 1-7 as the discussion) and part II as the conclusion. There are too many keywords, should be simplified.

The paragraphs have been broken up as suggested. The keywords have also been simplified and reduced.

  1. Data in figure 2 showed that as MH concentration increased, the integrity of pore structure decreased, why? What is the underlying reasons for that?

The MH scaffolds were fabricated such that MH was incorporated into the solution prior to crosslinking. Therefore, following crosslinking and freezing, the MH is directly incorporated within the polymer chain struts. This results in multiple changes in both the ice crystal formation and shape of the polymer chains forming the pores, directly leading to decreased integrity of the pore structure. Additionally, the hydrophilic/hydroscopic nature of the honey directly pulls and attracts the water with the distribution of honey throughout the polymer struts. As the polymer chains are disrupted by the honey, this results in an angular geometry and generally smaller pore sizes. This information has been updated in the discussion section.

  1. Also, why cryogel pores lost the circular structure required for cell infiltration and, instead, displayed angular pore geometry, fewer pores, and a larger number of pore sizes falling outside of the optimal range for cell infiltration?

Please see the answer to question 4 above.

  1. In figure 7-10, authors using the absorbance 550 nm to indicate the bacterial growth and biofilm formation, which is not intuitionistic to illustrate the antibacterial properties of the hydrogels and cyrogels. Should they changed into antibacterial rations?

While we appreciate the suggestion, the use of absorbance to quantify Crystal Violet stain is standard practice. We have added references to further support this remaining in the manuscript.

  1. Several related papers on this area can be included: https://doi.org/10.3390/gels5020021; https://doi.org/10.1016/j.ijbiomac.2023.125754.

Thank you for the suggestion! The first manuscript “Investigating Manuka Honey Antibacterial Properties When Incorporated into Cryogel, Hydrogel, and Electrospun Tissue Engineering Scaffolds” is already cited. We are choosing not to include “Highly absorbent bio-sponge based on carboxymethyl chitosan/poly-γ-glutamic acid/platelet-rich plasma for hemostasis and wound healing,” as this paper does not include any data using honey or focused on bacterial clearance/biofilm formation, the subject of this manuscript.

Reviewer 2 Report

Comments and Suggestions for Authors

No one doubts that the search for effective systems to combat bacterial infection is one of the most important tasks of medicinal chemistry. The authors correctly identified the problems and vector of development. The authors offer a very good and original solution to the problem through the development of manuka honey-doped chitosan-gelatin cryogel and hydrogel scaffolds. There are practically no similar works in the literature, so the novelty of the research is very high. The article is well illustrated, written competently and logically. The conclusion corresponds to the results of the work. The experiment was carried out methodically correctly. In addition, relevant literature references are provided. The abstract is clear, logical and reflects the essence of the work. This article will be of interest to many specialists and will be well cited. I recommend publishing after a minor revision. I kindly ask the authors to indicate the polydispersity index of chitosan, the average molecular weight and the method for its determination, the degree of deacetylation (this is important for the reproducibility of the results), and also to shorten the introduction, now it suffers from pronounced verbosity.

Author Response

Dear Editors and Reviewers,

We are pleased to submit the revised version of Gels (ID# gels-2652916): “Antibacterial Efficacy of Manuka Honey-Doped Chitosan-Gelatin Cryogel and Hydrogel Scaffolds in Reducing Infection.” We appreciate the reviewers' valuable feedback and have thoroughly addressed each of their concerns as detailed below. All revisions to the manuscript are tracked changes. Additionally, we have included a clean version of the manuscript with all changes accepted. We enhanced both the text and figures to enrich the paper; these modifications have notably enhanced the paper and improved its clarity.

Reviewer 2

No one doubts that the search for effective systems to combat bacterial infection is one of the most important tasks of medicinal chemistry. The authors correctly identified the problems and vector of development. The authors offer a very good and original solution to the problem through the development of manuka honey-doped chitosan-gelatin cryogel and hydrogel scaffolds. There are practically no similar works in the literature, so the novelty of the research is very high. The article is well illustrated, written competently and logically. The conclusion corresponds to the results of the work. The experiment was carried out methodically correctly. In addition, relevant literature references are provided. The abstract is clear, logical and reflects the essence of the work. This article will be of interest to many specialists and will be well cited. I recommend publishing after a minor revision. I kindly ask the authors to indicate the polydispersity index of chitosan, the average molecular weight and the method for its determination, the degree of deacetylation (this is important for the reproducibility of the results), and also to shorten the introduction, now it suffers from pronounced verbosity.

Thank you for your thoughtful review! Please see specific responses below:

  1. The polydispersity index of chitosan, the average molecular weight and the method for its determination, the degree of deacetylation (this is important for the reproducibility of the results).

The chitosan used in this study had an average molecular weight of 50,000-190,000 Da and a viscosity of 20-300 cP. Sigma Aldrich does not measure the polydispersity of this product and the molecular weight is determined only by viscosity. The physical form of this chitosan is 75-85% deacetylated. This information has been updated in the methods and the discussion now reflects the need for a future polydispersity test.

  1. Shorten the introduction, now it suffers from pronounced verbosity.

The introduction has been shortened to better articulate information in a concise manner.

Reviewer 3 Report

Comments and Suggestions for Authors

The present manuscript entitled “Antibacterial Efficacy of Manuka Honey-Doped Chitosan-Gelatin Cryogel and Hydrogel Scaffolds in Reducing Infection” by Mitchell et al., describes the testing varying concentrations of Manuka Honey (MH) in hydrogel and cryogel scaffolds. Scaffolds were evaluated to understand MH’s effects on physical properties and its antibacterial activity. Furthermore, the outcomes indicated that 5% MH cryogels and hydrogels appear to withstand physical property changes, maintaining desired porosity, swelling, and mechanics, while manifesting antibacterial and anti-biofilm activity. The authors report an interesting work. The objective and justification of the work are very clear. Therefore, I recommend it for publication. However, some minor issues are detailed below which need to be addressed before its final acceptance in Gels.

I advise the authors to take the following points into account while revising their manuscript.

Comment 1: There are some typographical and grammatical errors in the manuscript text, so the authors need to correct them in the revised manuscript. For e.g. Line 43, population [1]. [1]Despite these defense mechanisms should be population [1]. Despite these defense mechanisms; Line 113, biofilm formation.[17] should be biofilm formation [17].;

Comment 2: The Abstract needs to be revised, let the author focus main points, and explain the research question clearly also include the performed characterization techniques such as SEM details in the abstract, a slight revision of the abstract is required to attract a broad readership.

Comment 3: Figure 2 red color scale bar is not visible properly, so redraw the scale bar with a different color.

Comment 4: Manuka Honey (MH) (0%, 1%, 5%, 10%) incorporated into both chitosan-gelatin, why the authors selected only 0-10% of Honey concentration. Is there any specific reason clarify.

Comment 5: Why did the authors select 1% honey concentration instead of 2.5% concentration, they can double concentration ratios such as (0%, 2.5%, 5%, and 10%).

Comment 6: In Figure 6, bacterial names should be in italics.

Comment 7: The conclusion section should be moved after the Materials and Methods section to attain a broad readership.

Comment 8:  Include the conclusion section with clear quantitative findings and more emphasis on the findings and their implications may be mentioned in the conclusion section.

Comment 9: The homogeneity of the reference section needs to be maintained. In some references, journal names are in abbreviated form and some are in Full form. So please check and revise according to the journal's instructions.

Comments on the Quality of English Language

Minor editing of English language required.

Author Response

Dear Editors and Reviewer,

We are pleased to re-submit the revised version of Gels (ID# gels-2652916): “Antibacterial Efficacy of Manuka Honey-Doped Chitosan-Gelatin Cryogel and Hydrogel Scaffolds in Reducing Infection.” We appreciate the reviewer’s valuable feedback and have thoroughly addressed each of their concerns as detailed below. All revisions to the manuscript are tracked changes. Additionally, we have included a clean version of the manuscript with all changes accepted. We enhanced both the text and figures to enrich the paper; these modifications have notably enhanced the paper and improved its clarity.

Reviewer 3

The present manuscript entitled “Antibacterial Efficacy of Manuka Honey-Doped Chitosan-Gelatin Cryogel and Hydrogel Scaffolds in Reducing Infection” by Mitchell et al., describes the testing varying concentrations of Manuka Honey (MH) in hydrogel and cryogel scaffolds. Scaffolds were evaluated to understand MH’s effects on physical properties and its antibacterial activity. Furthermore, the outcomes indicated that 5% MH cryogels and hydrogels appear to withstand physical property changes, maintaining desired porosity, swelling, and mechanics, while manifesting antibacterial and anti-biofilm activity. The authors report an interesting work. The objective and justification of the work are very clear. Therefore, I recommend it for publication. However, some minor issues are detailed below which need to be addressed before its final acceptance in Gels.

Thank you for your thoughtful review.

I advise the authors to take the following points into account while revising their manuscript.

Comment 1: There are some typographical and grammatical errors in the manuscript text, so the authors need to correct them in the revised manuscript. For e.g. Line 43, population [1]. [1]Despite these defense mechanisms should be population [1]. Despite these defense mechanisms; Line 113, biofilm formation.[17] should be biofilm formation [17].;

We appreciate the reviewer catching these mistakes and have modified the text accordingly (along with a few other mistakes we noticed).

Comment 2: The Abstract needs to be revised, let the author focus main points, and explain the research question clearly also include the performed characterization techniques such as SEM details in the abstract, a slight revision of the abstract is required to attract a broad readership.

The abstract has been revised to reference all of the characterization techniques used in the study as suggested by the reviewer.

Comment 3: Figure 2 red color scale bar is not visible properly, so redraw the scale bar with a different color.

Thank you for noting this – scale bar/text color has been changed to white with a black background for improved visibility. This has also been modified in Figure 13.

Comment 4: Manuka Honey (MH) (0%, 1%, 5%, 10%) incorporated into both chitosan-gelatin, why the authors selected only 0-10% of Honey concentration. Is there any specific reason clarify.

We appreciate the authors comment. To clarify, these concentrations were based on previous work by our group and others. These percentages are ideal for inducing an antibacterial effect, without negatively impacting the chosen scaffold’s properties for tissue healing. We have updated this both within the Methods and Reference section.

Comment 5: Why did the authors select 1% honey concentration instead of 2.5% concentration, they can double concentration ratios such as (0%, 2.5%, 5%, and 10%).

Please see the answer to comment 4. All concentrations were based on previous studies conducted by our group and others. This has also been clarified within the text.

Comment 6: In Figure 6, bacterial names should be in italics.

This has been modified in Figure 6.

Comment 7: The conclusion section should be moved after the Materials and Methods section to attain a broad readership.

While we agree with the reviewer, this is based on the Gels template so we must leave the Conclusion section in this spot. If the editors approve the change, we are happy to move the section.

Comment 8:  Include the conclusion section with clear quantitative findings and more emphasis on the findings and their implications may be mentioned in the conclusion section.

The conclusion has been updated to include more quantitative information and an overview of the findings/implications, as suggested.

Comment 9: The homogeneity of the reference section needs to be maintained. In some references, journal names are in abbreviated form and some are in Full form. So please check and revise according to the journal's instructions.

The references have been modified.